# Postsurgical Thrombotic Microangiopathy and Deregulated Complement

**DOI:** 10.3390/jcm11092501

**Published:** 2022-04-29

**Authors:** Thijs T. W. van Herpt, Sjoerd A. M. E. G. Timmermans, Walther N. K. A. van Mook, Bas C. T. van Bussel, Iwan C. C. van der Horst, Jos G. Maessen, Ehsan Natour, Pieter van Paassen, Samuel Heuts

**Affiliations:** 1Department of Intensive Care Medicine, Maastricht University Medical Center+, 6229 HX Maastricht, The Netherlands; w.van.mook@mumc.nl (W.N.K.A.v.M.); bas.van.bussel@mumc.nl (B.C.T.v.B.); iwan.vander.horst@mumc.nl (I.C.C.v.d.H.); 2Department of Nephrology and Clinical Immunology, Maastricht University Medical Center+, 6229 HX Maastricht, The Netherlands; sjoerd.timmermans@mumc.nl (S.A.M.E.G.T.); p.van.paassen@mumc.nl (P.v.P.); 3Academy for Postgraduate Medical Training, Maastricht University Medical Center+, 6229 HX Maastricht, The Netherlands; 4Care and Public Health Research Institute (CAPHRI), Maastricht University, 6229 HX Maastricht, The Netherlands; 5Cardiovascular Research Institute Maastricht (CARIM), Maastricht University, 6229 HX Maastricht, The Netherlands; j.g.maessen@mumc.nl; 6Department of Cardiothoracic Surgery, Maastricht University Medical Center+, 6229 HX Maastricht, The Netherlands; e.natour@maastrichtuniversity.nl (E.N.); sam.heuts@mumc.nl (S.H.)

**Keywords:** thrombotic micro-angiopathy, aortic surgery, complement activation, thrombocytopenia, genetic variance, multidisciplinary approach, hemolytic uremic syndrome, complement inhibition

## Abstract

Postsurgical thrombotic microangiopathy (TMA) is a complication associated with significant morbidity and mortality. Still, the pathophysiological underlying mechanism of postsurgical TMA, a diagnosis often overlooked in postoperative patients with acute kidney injury and thrombocytopenia, is largely unknown. Here, we report the case of a 56-year-old male that developed anuric acute kidney injury, Coombs-negative hemolysis, and thrombocytopenia after surgical aortic arch replacement. Massive ex vivo complement activation on the endothelium, a rare complement gene variant in *C2*, at-risk haplotype *MCP*ggaac, and excellent response to therapeutic complement inhibition, points to the pivotal role of complement in the pathophysiology of disease. Moreover, the importance of a multidisciplinary team approach in (postsurgical) thrombocytopenia is emphasized.

## 1. Introduction

Aortic surgery, especially in a re-operative setting, carries a significant early morbidity and mortality rate, ranging between 5 and 10% [1]. Many complications are related to prolonged use of cardiopulmonary bypass (CPB), end-organ malperfusion and/or severe bleeding, either surgically related or secondary to acquired peri-operative coagulopathies. For example, thrombocytopenia, anemia, and/or acute kidney injury (AKI) are commonly encountered. Both postsurgical thrombocytopenia and anemia can be related to the surgical procedure, especially when performed with use of CPB, which induces hemodilution, hemolysis and consumption. In addition, AKI is a frequently observed after aortic arch surgery, as many of the open aortic procedures are performed during hypothermic circulatory arrest. Still, if this specific triad of consumptive thrombocytopenia, hemolytic anemia (often, Coombs negative), and AKI persists for several days following such extensive surgical procedures, further diagnostic evaluation is indicated. In fact, these patients might suffer from postsurgical thrombotic microangiopathy (TMA) [2], a severe and potentially life-threatening condition. 

Postsurgical TMA, although considered a rare entity, has been observed in up to 5% of patients following aortic surgery [3]. The pathogenic mechanism of postsurgical TMA remains to be established [4]. TMA reflects tissue responses to severe endothelial damage caused by distinct clinical conditions, comprising thrombotic thrombocytopenic purpura, infections, drug use, and deregulated complement (hereafter referred to as complement-mediated [C-] TMA), among others [5]. C-TMA, per definition, is a diagnosis of exclusion [6,7]. Deregulated complement is often caused by rare variants in complement genes that either regulate or activate complement and/or autoantibodies that that inhibit complement regulation [8,9]. Recent studies, however, have demonstrated deregulated complement as the catalyst of various TMAs presenting with coexisting conditions [5]. Thus, coexisting conditions, such as surgery, may lower the threshold for unrestrained complement activation and C-TMA to manifest. The approach to postsurgical TMA should therefore focus on the recognition of the “true” etiology in the earliest possible stage, having a major impact on treatment and prognosis [5]. In the current report, we demonstrate the pivotal role of complement activation in a patient with postsurgical TMA following aortic surgery.

## 2. Case Report

A 56-year-old male patient presented to our center with a residual dissection and progressive dilatation of the proximal descending aorta (68 mm) and other aortic segments (Figure 1A) following initial ascending aortic hemi-arch replacement during type A dissection four years earlier. Subsequently, in a re-operative setting, a total arch replacement using the frozen elephant trunk technique was performed during antegrade cerebral perfusion, with selective re-implantation of the supra-aortic vessels in the ascending aortic graft (Figure 1B demonstrates the imaging findings of the surgical result). During the following days, in the intensive care unit, the patient developed anuric AKI, requiring continuous veno-venous hemodialysis, thrombocytopenia (46 × 10^9^/L), and Coombs negative microangiopathic hemolytic anemia (hemoglobin, despite transfusion, 4.8 mmol/L; lactate dehydrogenase 1778 U/L, haptoglobin <0.10 g/L, and schistocytes 2+), suggesting TMA. Coagulation tests were within the (pseudo)normal range (activated partial thromboplastin time 28 s (reference value 23–32 s), prothrombin time 15.5 s (reference value 9.9–12.4 s)), while overt bleeding was absent. Plasmapheresis was initiated immediately, awaiting the results of enzymatic activity of von Willebrand factor cleaving protease (i.e., ADAMTS13), which proved to be within the normal range (49%; ref., >10%, excluding thrombotic thrombocytopenic purpura). Routine complement measures showed low C4 (0.07 g/L; reference range, 0.11–0.35 g/L) and C3 (0.43 g/L; reference range, 0.75–1.35 g/L). Moreover, serum induced massive ex vivo complement activation as determined by formation of the membrane attack complex (i.e., C5b9) on microvascular endothelial cells (364% compared to pooled normal human serum run in parallel (*p* value < 0.01)), indicating C-TMA [10]; other causes of TMA [5] were excluded. Despite five sessions of plasma exchange, hematologic indices and kidney function did not improve; thus, eculizumab (a monoclonal antibody that blocks C5 activation) was administered according to standard protocol [11], leading to a complete clinical remission during hospitalization (Figure 2). The patient was discharged to the surgical ward on the 30th postoperative day.

DNA sequencing and multiplex ligation-dependent probe amplification of complement genes (i.e., *CFH*, *CFI*, *CD46*, *CFB*, *C3*, *CFHR1*-*5*, *C2*, *CFP*, and *THBD*) demonstrated a rare variant in *C2* c.841_849+19del, with a minor allele frequency of 0% (Exome Variant Server) to 0.4% (gnomAD), at-risk haplotype *MCP*ggaac, and deletion of *CFHR1* and *CFHR3*; all the genetic abnormalities came in heterozygosity. Factor H autoantibodies, however, were not found.

## 3. Discussion

The current report describes a patient with postsurgical TMA following aortic surgery presenting with massive ex vivo complement activation, a novel variant in *C2*, and excellent response to therapeutic complement inhibition, pointing to complement activation as the driving factor of disease. Postsurgical TMA, indeed, has been linked to deregulated complement in a small subset of patients [4]. Thus, recognition of C-TMA is of the utmost clinical importance to guide treatment and follow-up. We therefore emphasize the need for a multidisciplinary team approach to initiate therapeutic complement inhibition in the earliest possible stage of disease (Figure 3) [12].

The diagnosis of (postsurgical) C-TMA can be challenging and is frequently delayed because features of TMA overlap with several more obvious sequelae of surgery [4]. For example, thrombocytopenia is often related to CPB, bleeding, and/or hemodilution, but may also indicate TMA, particularly when persistent and associated with microangiopathic hemolysis [13]. At any time, diffuse intravascular coagulation should be ruled out, although one must also consider that coagulation tests can be disrupted after extensive surgical procedures [14]. C-TMA should be considered in patients with a normal enzymatic activity of von Willebrand factor cleaving protease presenting with severe AKI not responding on conservative treatment but no apparent other cause of TMA [5]. Additionally, cognitive impairment within the first week following surgery is common [4,14]. As such, a multidisciplinary approach is warranted, allowing for a rapid and complete differential and etiology-directed treatment.

Most reports on postsurgical TMA do not provide data on complement measurements. To the best of our knowledge, two patients with postsurgical TMA presented with variants of uncertain significance in complement genes [4], suggesting C-TMA. Of note, open but not endovascular aortic surgery has been linked to systemic complement activation via the classical and/or lectin pathway [15]; this phenomenon relates to CPB [15,16] and/or ischemia reperfusion injury. Both pathways converge to C3 after the activation of C4 and *C2* to form the C3 convertase (i.e., C4bC2b). Subsequently, the activation of C3 leads to the assembly of the C5 convertase, activating the terminal complement pathway via C5 and the formation of C5b9. Moreover, prosthesis-related hemolysis with the release of heme can activate complement on the endothelium [17,18]. At presentation, our patient’s serum induced massive ex vivo C5b9 formation on the endothelium, reflecting the activation of the terminal complement pathway identical to serum from patients with C-TMA [8,19]. DNA sequencing revealed a novel variant in *C2*; recently, a gain-of-function *C2* protein, a paralog of the alternative pathway protein complement factor B [20], that leads to unrestrained complement activation via the classical and/or lectin pathway has been demonstrated in C-TMA. The combination of low levels of C4 and C3, indeed, confirmed involvement of the classical and/or lectin pathway. Moreover, *MCP*ggaac decreases transcriptional activity of the *CD46* promoter region, affecting complement regulation downstream of *C2* [21]. We therefore hypothesize that the surgical procedure lowered the threshold for unrestrained complement activation and, thus, for C-TMA to manifest. 

Postsurgical TMA likely represents a mixture of distinct causes, only some of which may be associated with deregulated complement. Plasma exchange should be started immediately in patients with postsurgical TMA. Plasma exchange has been associated with clinical remission in 30% to 80% [3,4,14], pointing to a plasmatic imbalance in many patients with postsurgical TMA. Patients not responding to plasma exchange, however, may have C-TMA and should therefore be treated with therapeutic complement inhibition such as eculizumab. Therapeutic complement inhibition has dramatically changed the outcome of patients with C-TMA; patient and kidney survival improved from less than half of untreated patients [9] to ~90% at 12 months [11,22]. Future studies should therefore focus on the efficacy of therapeutic complement inhibition in patients with postsurgical TMA and, in particular, those with deregulated complement. Many such patients present with severe AKI and massive ex vivo C5b9 formation on the endothelium [10], identical to our patient. In our experience, a multidisciplinary team approach improves the diagnosis and treatment of patients with TMA, assuring targeted treatment at the earliest possible stage of disease. Restrictive use of therapeutic complement inhibition may be feasible as the coexisting condition is present for a short time, lowering the very high costs of treatment [23]. The optimal duration of treatment, however, remains to be established. Furthermore, long-term outcome data of C-TMA associated with *C2* variants are warranted.

In conclusion, we demonstrate the pivotal role of complement activation in a patient with postsurgical TMA following aortic surgery as indicated by: (i) massive ex vivo C5b9 formation on the endothelium, (ii) a novel *C2* variant combined with *MCP*ggaac, and (iii) excellent response upon eculizumab. Furthermore, we emphasize a multidisciplinary approach of (postsurgical) TMA after cardiovascular surgery to optimize treatment for this life-threatening complication (Figure 3).

## Figures and Tables

**Figure 1 jcm-11-02501-f001:**
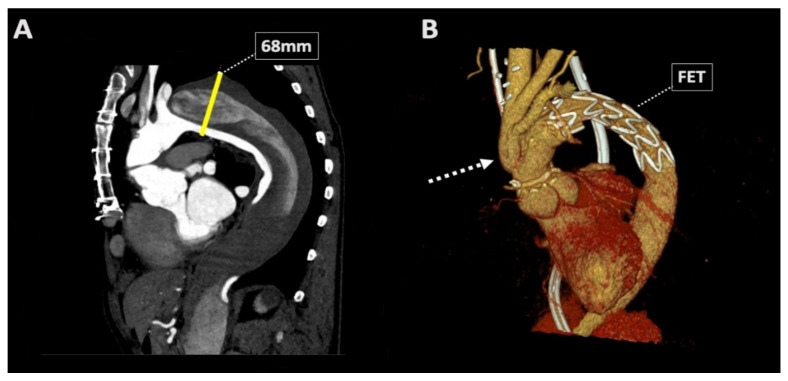
Pre- and postoperative contrast-enhanced CT scans. (**A**) Pre-operative CT demonstrating the 68 mm descending aortic aneurysm as the surgical indication; (**B**) three-dimensional reconstruction of postoperative CT findings, showing resection of the aneurysm using a total arch replacement technique (FET) with debranching of the supra-aortic vessels. CT: computed tomography, FET: frozen elephant trunk.

**Figure 2 jcm-11-02501-f002:**
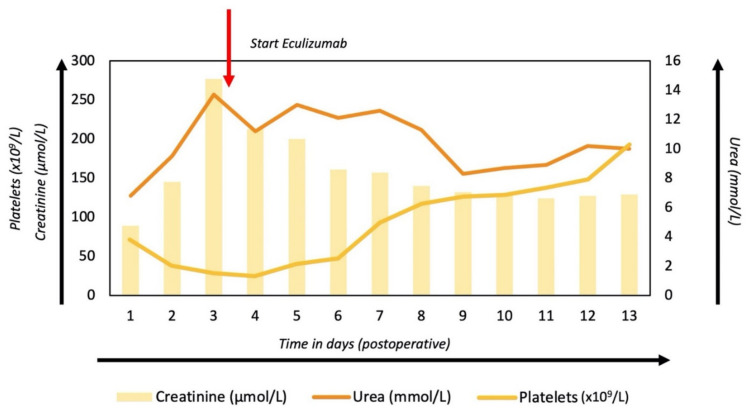
Postoperative course of kidney function.

**Figure 3 jcm-11-02501-f003:**
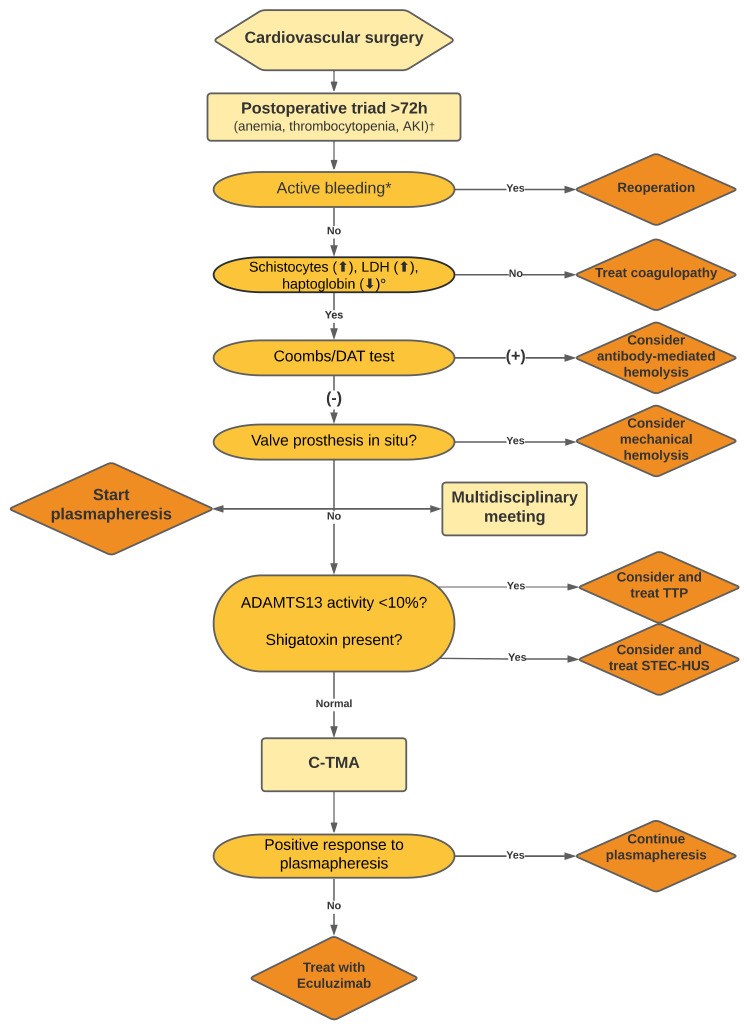
Proposal for standardized assessment of thrombocytopenia, anemia, and AKI after cardiovascular surgery. The combination of (consumptive) thrombocytopenia, microangiopathic hemolytic anemia (often, DAT negative), and AKI suggests TMA and thus, testing for the enzymatic activity of ADAMTS13 and Shiga toxin-producing *E. coli* infection should be requested simultaneously. AKI: acute kidney injury, C-TMA: complement mediated thrombotic microangiopathy, DAT: direct antiglobulin test, LDH: lactate dehydrogenase, STEC-HUS: Shiga toxin-producing Escherichia *coli*-associated hemolytic uremic syndrome, TTP: thrombotic thrombocytopenic purpura. ^†^ Anemia is defined as hemoglobin <5.0 mmol/L, thrombocytopenia defined as count below 150 × 10^9^/L, AKI defined as anuria, requirement for continuous veno-venous hemodialysis and/or triplication of pre-existing creatinine concentration. * Active bleeding is defined as >50 mL/h thoracic drain production after the first 48 h following surgery. ° Increased LDH defined as concentration >600 U/L, decreased haptoglobin defined as concentration <0.20 g/L. Please note that this is an expert opinion provided by the authors not necessarily reflected in literature.

## Data Availability

The data presented in this study are available on request from the corresponding author.

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
