# Peer review of "Postsurgical Thrombotic Microangiopathy and Deregulated Complement"

_jcm, 2022, doi:10.3390/jcm11092501_

Round 1

Reviewer 1 Report

Thank the authors so much for presenting very interesting case with a new variant of C2.  A proposal for assessment in Figure 3 is insightful.  Due to proper diagnosis and treatment, the clinical course of the patient was fine, great.

Only one comment, in the 3rd paragraph in discussion, “We hypothesize that the surgical procedure lowered the threshold for complement activation,,,”. I agree this concept. However, “surgical procedure” includes too many, various things, anesthesia, stent itself, patients condition,,,.  Currently, what do you have any idea, what appears at least related to TMA in surgical procedure? Please give any comments.

Author Response

Rebuttal reviewer #1:

First of all, we would like to express our gratitude for the valuable comment of the reviewer. Cardiovascular surgery and, in particular, open aortic aneurysm repair, has been associated with systemic complement activation via the classical and/or lectin pathway [1]. This phenomenon seems to be associated with cardiopulmonary bypass [1,2] and/or ischemia reperfusion injury. Moreover, prosthesis related hemolysis with release of heme can activate complement on the endothelium [3,4]. Our patient, indeed, presented with low levels of circulating C4 and C3, confirming complement activation via the classical and/or lectin pathway; the C2 variant points to the genetic risk for C-TMA to occur. We presume that the aforementioned conditions (i.e., CPB, ischemia reperfusion injury, and heme) lowered the threshold for unrestrained complement activation and thus, TMA to manifest.

According to the comment of the reviewer, we changed the text into: “Of note, open but not endovascular aortic surgery has been linked to complement activation via the classical and/or lectin pathway; this phenomenon relates to CPB and/or ischemia reperfusion injury. (...) Moreover, prosthesis related hemolysis with release of heme can activate complement on the endothelium.

  1. Fiane, A.E.; Videm, V.; Lingaas, P.S.; Heggelund, L.; Nielsen, E.W.; Geiran, O.R.; Fung, M.; Mollnes, T.E. Mechanism of Complement Activation and Its Role in the Inflammatory Response After Thoracoabdominal Aortic Aneurysm Repair. Circulation 2003, 108, 849–856, doi:10.1161/01.CIR.0000084550.16565.01.
  2. Hoel, T.N.; Videm, V.; Mollnes, T.E.; Saatvedt, K.; Brosstad, F.; Fiane, A.E.; Fosse, E.; Svennevig, J.L. Off-Pump Cardiac Surgery Abolishes Complement Activation. Perfusion 2007, 22, 251–256, doi:10.1177/0267659107084142.
  3. Frimat, M.; Tabarin, F.; Dimitrov, J.D.; Poitou, C.; Halbwachs-Mecarelli, L.; Fremeaux-Bacchi, V.; Roumenina, L.T. Complement Activation by Heme as a Secondary Hit for Atypical Hemolytic Uremic Syndrome. Blood 2013, 122, 282–292, doi:10.1182/blood-2013-03-489245.
  4. May, O.; Merle, N.S.; Grunenwald, A.; Gnemmi, V.; Leon, J.; Payet, C.; Robe-Rybkine, T.; Paule, R.; Delguste, F.; Satchell, S.C.; et al. Heme Drives Susceptibility of Glomerular Endothelium to Complement Overactivation Due to Inefficient Upregulation of Heme Oxygenase-1. Frontiers in Immunology 2018, 9.

Author Response

Rebuttal reviewer #1:

Thank the authors so much for presenting very interesting case with a new variant of C2.  A proposal for assessment in Figure 3 is insightful.  Due to proper diagnosis and treatment, the clinical course of the patient was fine, great.

Only one comment, in the 3rd paragraph in discussion, “We hypothesize that the surgical procedure lowered the threshold for complement activation”. I agree this concept. However, “surgical procedure” includes too many, various things, anesthesia, stent itself, patients condition.  Currently, what do you have any idea, what appears at least related to TMA in surgical procedure? Please give any comments.

First of all, we would like to express our gratitude for the valuable comment of the reviewer. Cardiovascular surgery and, in particular, open aortic aneurysm repair, has been associated with systemic complement activation via the classical and/or lectin pathway [1]. This phenomenon seems to be associated with cardiopulmonary bypass [1,2] and/or ischemia reperfusion injury. Moreover, prosthesis related hemolysis with release of heme can activate complement on the endothelium [3,4]. Our patient, indeed, presented with low levels of circulating C4 and C3, confirming complement activation via the classical and/or lectin pathway; the C2 variant points to the genetic risk for C-TMA to occur. We presume that the aforementioned conditions (i.e., CPB, ischemia reperfusion injury, and heme) lowered the threshold for unrestrained complement activation and thus, TMA to manifest.

According to the comment of the reviewer, we changed the text into: “Of note, open but not endovascular aortic surgery has been linked to complement activation via the classical and/or lectin pathway; this phenomenon relates to CPB and/or ischemia reperfusion injury. (...) Moreover, prosthesis related hemolysis with release of heme can activate complement on the endothelium.

Reviewer 2 Report

Dear Authors,
I found this case report interesting and well-written.
I have no comments on the description of the case and general discussion, whereas I have some remarks on Figure 3 and your proposal for a standardized assessment of signs of postoperative TMA.
Indeed, I think that it should be underlined (also changing the figure organization) that in the occurrence of signs of TMA, assessment of index of hemolysis and coagulative status, ADAMTS13 activity, and the presence of Shiga-toxin should be simultaneous and not sequential (see, for example 
Tsai HM. A mechanistic approach to the diagnosis and management of atypical hemolytic uremic syndrome. Transfus Med Rev. 2014 Oct;28(4):187-97; Sridharan M, et al. Go Atypical hemolytic uremic syndrome: Review of clinical presentation, diagnosis, and management. J Immunol Methods. 2018 Oct;461:15-22). This aspect is also important to highlight that the timely initiation of treatment is essential to achieving clinical remission.
Moreover, regarding the treatment, I think that the current view is that in the case of complement-mediated TMA a therapy based on complement inhibition (namely, eculizumab) is the first choice, while plasma therapy should be restricted to specific cases, such unavailability of eculizumab or cases of aHUS associated with anti- FH-positive antibodies (see, Goodship TH, Cook HT, Fakhouri F, Fervenza FC, Frémeaux-Bacchi V, Kavanagh D, et al.  Atypical hemolytic uremic syndrome and C3 glomerulopathy: conclusions from a "Kidney Disease: Improving Global Outcomes" (KDIGO) Controversies Conference. Kidney Int. 2017 Mar;91(3):539-551). This point should be clearly discussed.

Minor observation:
Please revise this sentence in the abstract: "Massive ex vivo complement activation on the endothelium, a rare complement gene variant in C2, at-risk haplotype MCPggaac, and excellent response to therapeutic complement inhibition, pointing to the pivotal role of complement in the pathophysiology of disease".
It misses a verb.

Author Response

I found this case report interesting and well-written. I have no comments on the description of the case and general discussion, whereas I have some remarks on Figure 3 and your proposal for a standardized assessment of signs of postoperative TMA.
Indeed, I think that it should be underlined (also changing the figure organization) that in the occurrence of signs of TMA, assessment of index of hemolysis and coagulative status, ADAMTS13 activity, and the presence of Shiga-toxin should be simultaneous and not sequential (see, for example Tsai HM. A mechanistic approach to the diagnosis and management of atypical hemolytic uremic syndrome. Transfus Med Rev. 2014 Oct;28(4):187-97; Sridharan M, et al. Go Atypical hemolytic uremic syndrome: Review of clinical presentation, diagnosis, and management. J Immunol Methods. 2018 Oct;461:15-22). This aspect is also important to highlight that the timely initiation of treatment is essential to achieving clinical remission.

Authors: We thank the reviewer for this important question and agree that the previous version of the Figure might lead to a delay in diagnosis. It is important to stress that the proposed flowchart applies to a specific group of patients, that is, patients undergoing cardiovascular surgery, as stated at the top part of the flowchart. The triad of anemia, thrombocytopenia, and acute kidney injury, although likely caused by “common” complications of cardiothoracic surgery (e.g., anastomic bleeding, coagulopathies, prosthesis related hemolysis), suggests TMA. Thus, we agree with the reviewer that tests for ADAMTS13’s enzymatic activity and Shiga toxin-producing E. coli infection should be ordered simultaneously to prevent a diagnostic delay.

According to the comment of the reviewer, we updated Figure 3. Also, we underlined that the flowchart only applies to patients with suspected TMA after cardiovascular surgery, both in Figure 3 and main text.

Moreover, regarding the treatment, I think that the current view is that in the case of complement-mediated TMA a therapy based on complement inhibition (namely, eculizumab) is the first choice, while plasma therapy should be restricted to specific cases, such unavailability of eculizumab or cases of aHUS associated with anti- FH-positive antibodies (see, Goodship TH, Cook HT, Fakhouri F, Fervenza FC, Frémeaux-Bacchi V, Kavanagh D, et al.  Atypical hemolytic uremic syndrome and C3 glomerulopathy: conclusions from a "Kidney Disease: Improving Global Outcomes" (KDIGO) Controversies Conference. Kidney Int. 2017 Mar;91(3):539-551). This point should be clearly discussed.

Authors: We agree with the reviewer that therapeutic complement inhibition should be started at the earliest possible stage of disease in patients with C-TMA. However, only a subset of patients with postsurgical TMA present with TMA on the background of complement dysregulation. We propose “first line” plasma exchange because a clinical response can be expected in up to 80% cases [1–3]. C-TMA is common in patients with severe acute kidney injury and/or massive ex vivo C5b9 formation on the endothelium [4]. Many of such patients do not respond to plasma exchange. We therefore state that: “Patients not responding to plasma exchange may have C-TMA and should therefore be treated with therapeutic complement inhibition, such as, eculizumab. (...) Many of such patients present with severe AKI and massive ex vivo C5b9 formation on the endothelium, identical to our patient.

Minor observation:
Please revise this sentence in the abstract: "Massive ex vivo complement activation on the endothelium, a rare complement gene variant in C2, at-risk haplotype MCPggaac, and excellent response to therapeutic complement inhibition, pointing to the pivotal role of complement in the pathophysiology of disease".
It misses a verb.

Authors: According to the comment of the reviewer, the text has been changed into: “Massive ex vivo complement activation on the endothelium, a rare complement gene variant in C2, at-risk haplotype MCPggaac, and excellent response to therapeutic complement inhibition points to the pivotal role of complement in the pathophysiology of disease.

Round 2

Reviewer 2 Report

Dear Authors, I think that the paper is now more clear. No further comments by me.